# Anthocyanin-Rich Vegetables for Human Consumption—Focus on Potato, Sweetpotato and Tomato

**DOI:** 10.3390/ijms23052634

**Published:** 2022-02-27

**Authors:** Autar K. Mattoo, Sangam L. Dwivedi, Som Dutt, Brajesh Singh, Monika Garg, Rodomiro Ortiz

**Affiliations:** 1USDA-ARS, Sustainable Agricultural Systems Laboratory, Beltsville, MD 20705-2350, USA; autar.mattoo@usda.gov; 2Independent Researcher, Hyderabad 500016, India; sangam375@gmail.com; 3Central Potato Research Institute, Shimla 171001, India; sd_bio@yahoo.com (S.D.); birju16@gmail.com (B.S.); 4National Agri-Food Biotechnology Institute, Mohali 160071, India; monikagarg@nabi.res.in; 5Department of Plant Breeding, Swedish University of Agricultural Sciences, Sundsvagen, 10 Box 190, SE 23422 Lomma, Sweden

**Keywords:** biofortification, biosafety regulations, colored vegetables, flavonoids, gene editing, genetic tags, transgenes

## Abstract

Malnutrition, unhealthy diets, and lifestyle changes have become major risk factors for non-communicable diseases while adversely impacting economic growth and sustainable development. Anthocyanins, a group of flavonoids that are rich in fruits and vegetables, contribute positively to human health. This review focuses on genetic variation harnessed through crossbreeding and biotechnology-led approaches for developing anthocyanins-rich fruit and vegetable crops. Significant progress has been made in identifying genes involved in anthocyanin biosynthesis in various crops. Thus, the use of genetics has led to the development and release of anthocyanin-rich potato and sweet potato cultivars in Europe and the USA. The purple potato ’Kufri Neelkanth’ has been released for cultivation in northern India. In Europe, the anthocyanin-rich tomato cultivar ‘Sun Black’ developed via the introgression of *Aft* and *atv* genes has been released. The development of anthocyanin-rich food crops without any significant yield penalty has been due to the use of genetic engineering involving specific transcription factors or gene editing. Anthocyanin-rich food ingredients have the potential of being more nutritious than those devoid of anthocyanins. The inclusion of anthocyanins as a target characteristic in breeding programs can ensure the development of cultivars to meet the nutritional needs for human consumption in the developing world.

## 1. Introduction

Anthocyanins (ACNs) and anthocyanidins are plant pigments, with flavylium (2-phenylchromenylium) ions being their basic core structure. ACNs are water-soluble, and present as a vacuolar pigment with a range of diverse colors (red, purple, black, or blue), while anthocyanidin is the sugar-free counterpart of anthocyanin. ACNs are the largest group of water-soluble pigments, located in the plant cell vacuoles. They are esterified with one or more acid groups, as acylated anthocyanins, and are characterized as acylated or nonacylated anthocyanins based on the presence or absence of the acyl group. Acylated anthocyanins constitute a variety of forms based on the number of acyl groups (mono-, di-, tri-, and tetra-acylated anthocyanins). Differences in pH, light, temperature, and structure influence the color and stability of ACNs. They appear red in acidic conditions but turn blue when the pH increases [1,2]. Acylated ACNs show higher stability and higher antioxidant activity than nonacylated types [3]. Fruits and vegetables (FV) contain both acylated and nonacylated ACN forms [2].

Fruits and vegetables with different colors provide a range of nutrients and bioactive compounds including phytochemicals, vitamins, minerals, and fibers [4]. ACNs have promising health-promoting groups of phytochemicals. Several reports involving double-blind, randomized, and control trials have revealed dose-dependent effects of ACNs on chronic diseases, such as on glucose and lipid metabolism, weight management, plasma lipids and cholesterol efflux, or on platelet functions and cholesterol efflux in subjects with dyslipidemia [5,6,7,8,9]. Detecting either parent ACNs or their metabolites in tissue, as evidenced in mice kidney, liver, heart, and lung and pig brain tissues, could provide an evidentiary link between tissue ACNs and their associated health effects [10].

A two-prospective cohort and meta-analysis involving US men and women in 26 revealed nonlinear inverse associations of FV intake with total mortality and cause-specific mortality. Intake of ≈5 servings per day of FV was associated with the lowest mortality, while intake above this level was not associated with additional risk reduction. Daily intake of 5 servings relative to the reference level (2 servings per day) was associated with hazard ratios (95% CI) of 0.87 (0.85–0.90) for total mortality, 0.88 (0.83–0.94) for CVD mortality, 0.90 (0.86–0.95) for cancer mortality, and 0.65 (0.59–0.72) for respiratory disease mortality, with similar results (summary risk ratio of mortality for 5 servings per day was =0.87 with a 95% CI in the 0.85–0.88; *P*_nonlinear_ < 0.001) based on dose-response meta-analysis that included 145,015 deaths accrued in 1,892,885 subjects. Thus, higher intakes of most subgroups of FV were associated with lower mortality [11].

Of late, several scientific articles, including original papers and reviews [12,13,14,15,16,17,18] have been published about the beneficial effects of anthocyanin on human health. There is thus an interest in such bioactive compounds, not only from a pharmaceutical standpoint but also from a nutraceutical perspective. Moreover, the use of biostimulants, mycorrhiza, and semi-natural compounds such as melatonin and phytomelatonin have been used to increase anthocyanin content in food crops, including FV [19,20,21].

In this review, we focus on the anthocyanins with specific reference to their useful variable germplasm, genetics, and molecular basis of inheritance. We also present data on functional and candidate genes as well as DNA markers associated with anthocyanins together with progress toward developing anthocyanins rich potato [*Solanum tuberosum*], sweet potato [*Ipomoea batatas*], and tomato [*Solanum lycopersicum*]) cultivars, which are the three most consumed vegetables, via crossbreeding and biotechnology.

## 2. Natural Variation and Diversity in Gene Pools

Germplasm is the main source for crop improvement programs because it represents a large pool of genetic diversity from which researchers can draw allelic variation for further use to develop, for instance, cultivars with significant nutritional and health benefits.

### 2.1. High Throughput Assay

Ultrasound-assisted extraction (UE) is an advanced green, fast, and ecological extraction technique to generate high-quality extracts from natural products with high precision (coefficient of variation less than 5%) and reproducibility. The optimal extraction time determined for total phenols and total anthocyanins were 15- and 5-min, respectively. Temperature and percent methanol are the most influential variables. UE has been evaluated in determining total phenols and anthocyanins in berry crops, grape, and potato [22,23,24,25]. It is an ideal assay for quantifying total phenol and anthocyanin contents in plants. Interestingly, in sweet potato, the root-flesh color (red-purple, deep purple to white purplish) and their lightness values (L *) were negatively correlated (*R*^2^ = 0.65) with anthocyanin content [26]. Indirectly, this specified that deep-colored root-flesh contains more anthocyanins. Thus, intensity of color (more intense color, more anthocyanins) is a marker to identify high anthocyanins segregants in early generation breeding populations followed by wet chemistry tests on advanced breeding phenotypically uniform germplasm at later generations.

### 2.2. Genotype × Environment Interactions (GEI)

Differences in geographical locations due to variation in light intensity, photoperiod, temperature, soil fertility, stress levels, stage of tuber or storage root formation, and harvest all influence anthocyanin contents in potato and sweet potato. High temperature was found to decrease anthocyanin levels in purple-skinned and fleshed potato by 48% [27]. Thus, potatoes harvested in cooler climates produced 1.24 times more anthocyanins than those in warmer climates [28]. Potatoes grown under long days (14–15 h) and cooler temperatures accumulate about 2.5 times higher anthocyanin levels than those grown under short days (12–14 h) and higher temperatures [29]. Staggered harvest can affect anthocyanin content in potato due to variation in global solar irradiation (r < 0.6252) [30]. Both growth environment (E) and genotype (G) can cause significant (*p* < 0.01) differences in total anthocyanin content, with greater variation due to G rather than E in potato [31].

Overall, G, E, and GEI affect the content and composition of anthocyanins. Thus, multi-environment testing across diverse agro-ecologies has been recommended to identify stable and anthocyanin-rich germplasm for use in crop breeding as well as identifying sites more favorable to the production of anthocyanin-rich crops.

### 2.3. Genetic Variation

Fruits and vegetables in general contain higher amounts of anthocyanin compounds with more in fruits than in vegetables. The total anthocyanin content in FV may vary considerably among different plant genera, species and cultivars, and could be strongly influenced by light, temperature and agronomic factors. Phenol Explorer Online Database is a good source of information about anthocyanins in the plant kingdom. Euclidean distance-based cluster analysis of anthocyanin data grouped the species into 14 distinct clusters (< 15 mg 100 g^−1^; > 15 mg 100 g^−1^; < 100 mg 100 g^−1^; > 100 mg 100 g^−1^; < 130 mg 100 g^−1^; > 130 mg 100 g^−1^; < 150 mg 100 g^−1^; > 150 mg 100 g^−1^; > 275 mg 100 g^−1^; > 275 mg 100 g^−1^; < 300 mg 100 g^−1^; > 300 mg 100 g^−1^; < 500 mg 100 g^−1^; > 500 mg 100 g^−1^), with grape (*Vitis vinifera*), date (*Phoenix dactylifera*), and fig (*Ficus carica*) at a lower side (< 15 mg 100 g^−1^), while berry and cherry crops are on the higher side (150–500 mg 100 g^−1^). Purple and red potatoes have anthocyanin levels of ~15 mg 100 g^−1^ or < 15 mg 100 g^−1^, respectively [19].

Differences in anthocyanin content that varied from 4-, 7-, 9-, and 95-fold in flesh and 9-fold in skin, were found among Andean potato germplasm [32,33,34,35,36]. Five anthocyanidins namely, delphinidin, cyanidin, petunidin, pelargonidin, and peonidin were quantified in tubers from 109 highly divergent diploid germplasm and found to differ in skin and flesh color, from yellow to purple, in the Colombian germplasm of *Solanum tuberosum* L. Group Phureja. Principal component analysis (PCA) clustered the accessions into six different groups with a majority of accessions in Group 1 represented by lowest concentrations of five anthocyanin compounds; group 2 had considerable variation in pelargonidin (17–38 mg 100 g^−1^ dry matter, DM); group 3 was represented by petunidin (19–76 mg 100 g^−1^ DM) and peonidin (up to 23 mg 100 g^−1^ DM); group 4 included pelargonidin (64–85 mg 100 g^−1^ DM) and petunidin (up to 28 mg 100 g^−1^ DM); group 5 included accessions with highest content of pelargonidin (87–168 mg 100 g^−1^ DM) but had the lowest content of the remaining four anthocyanins; while group 6 represented accessions with lowest pelargonidin content but with higher concentrations of other four anthocyanins [37].

In the red- and purple-potato Korean cultivars and breeding clones, 26 anthocyanins were found, of which 24 were enriched in pelanin, peonanin, and petanin as principal anthocyanins while the red and purple potatoes grouped into distinct clusters [35]. Differences in anthocyanin patterns were also noted among native Andean potato germplasm. For example, red accessions contained predominantly pelargonidin derivatives, while the purple type had petunidin as a major anthocyanidin [32,33]. About 33% (660 cultivated and 1960 wild accessions) of the 6100-potato germplasm housed in the genebank at The Leibniz Institute of Plant Genetics and Crop Plant Research (IPK; http://glks.ipk-gatersleben.de/home.php, accessed on 22 January 2022) are highly variable anthocyanin accessions in color patterns (partial to complete; peel and flesh containing the same color or colored peel and noncolored flesh; degree of pigmentation either spots, stripes, or rings). At five locations in the Czech Republic the total anthocyanins content included 16 red- and purple-fleshed potatoes ranging from 0.7 mg 100 g^−1^ FW (cv. ‘Blue Congo’) to 74.3 mg 100 g^−1^ FW (cv. ‘Blaue Ludiano’). ‘Highland Burgundy Red’ had a high proportion of pelargonidin (98.7%), whereas ‘British Columbia Blue’ contained almost exclusively cyanidin. ‘Violette’ and ‘Vitelotte’ had a relatively high content of malvidin, while ‘Shetland Black’ had a higher content of peonidin (on average 36.7%). Petunidin was found abundant in cultivars ‘Valfi’, ‘Blue Congo’, ‘Salad Blue’, ‘Blaue St’. ‘Galler’, ‘Blaue Hindel Bank’, ‘Blaue Ludiano’, ‘Blaue Schweden’, ‘Farbe Kartoffel’ and ‘Salad Red’ [38]. Multi-fold differences in anthocyanins were reported among 26 potato cultivars differing in skin and flesh color that were evaluated for two seasons in New Zealand [39]. The total anthocyanin content in potato germplasm does not reach the level of anthocyanins reported in berry crops [19].

The structure and content of the secondary metabolites are likely to be affected by cooking because of high temperature (~100 °C). However, cooking of pigmented potatoes still provided valuable levels of phenolic compounds and antioxidant activity independent of temperatures [40,41].

CIP genebank include 16 purple-fleshed sweet potato (PFSP) accessions as sources of root anthocyanins (https://www.genesys-pgr.org/subsets/740f8b1e-8e3e-4773-a923-3e84eda56932, accessed on 22 January 2022). Six-fold differences in anthocyanin contents among purple-fleshed sweet potato germplasm have been recorded [42].

In comparison to potato cultivars, tomato fruit is deficient in secondary metabolites such as flavonoids and anthocyanins [43], therefore its enrichment for these metabolites has been pursued through genetic engineering in recent years [44,45,46,47,48]. Nonetheless, wild tomatoes such as *Solanum chilense, S. habrochaites, S. cheesmanii*, and *S. lycopersicoides*, unlike the cultivated tomato, do produce anthocyanins in the sub-epidermal fruit tissue [49,50]. VIR *Lycopersicon* collection includes 7678 accessions of one cultivated and nine wild species [51]. A comparative assessment of 70 tomato accessions with different fruit colors unfolded substantial variation for biochemical composition, including anthocyanins. The anthocyanins in cultivated tomato with purple-red and yellow-purple fruits, respectively, ranged from 32.89 to 588.86 and 87.91 to 161.22 mg^−1^ 100 g, while in wild tomato fruits the anthocyanin levels varied from 84.31 to 152.71 mg^−1^ 100 g. Among cultivars, ‘Ananas noire’ and ‘Indigo Clakamas Blue Berry’ had exceptionally high anthocyanins: 430.30 ± 98.35 and 588.86 ± 171.89 mg^−1^ 100 g, respectively. A dendrogram based on cluster analysis of biochemical parameters grouped the accessions into six clusters, where the first two clusters characterized accessions with high content of anthocyanins and chlorophylls and ascorbic acid in fruits [52]. A few VIR “*Lycopersicon*” accessions, unlike potato germplasm, had total anthocyanin in the range found for berry crops [19].

Genetic engineering involving specific transcription factors led to the enrichment of anthocyanins at concentrations that averaged 2.83 ± 0.46 mg g^−1^ fresh weight [45]. Likewise, the cultivar ‘Sun Black’ (SB) developed by crossing an *Anthocyanin fruit* introgression line (*Aft*, LA1996) with *atroviolaceum* line (*atv*, LA0797) led to anthocyanin content similar to that found in anthocyanin-rich eggplant (*Solanum melongena*), red lettuce (*Lactuca sativa*), light-colored strawberry (*Fragaria*
*× ananassa*), and cherry (*Prunus cerasus*) [48]. This research has demonstrated the power of genetic engineering as an important asset to develop anthocyanins-rich variants where crossbreeding programs are unable to deliver such a desired germplasm.

High anthocyanin content is associated with a bitter taste. However, the steamed tubers of two deep purple genotypes (MSU10010-43, ‘Antin 3′) and one white-purplish genotype (MSU1000115) showed differences in texture and taste [26]. Genotypic differences among purple-fleshed sweet potato genotypes were also reported for developing nutritious food products with respect to their physical properties (yield, lightness, hardness) and sensory attributes (color, aroma, texture, taste). For example, sweet potato genotypes with purple flesh in combination with yellow/orange or white color (MSU 06046-74, ‘Antin 1′, MSU 06044-05) were suitable for making crispy chips followed by those with medium purple color (‘Aya Murasaki’, MIS 0612-73, MIS 0614-02, MIS 0601-22, MSU 06014-51). Conversely, sweet potato with deep purple color (MSU 06028-71, MSU 06046-48) was more suitable for noodles [42]. Such attributes may be exploited in breeding program to develop product-based sweet potatoes.

Genotypic differences in the accessibility and biotransformation of anthocyanins among cooked PFSPs were revealed via a dynamic in vitro human gastrointestinal (GI) model that stimulated gut digestive conditions, with accession-dependent variation in anthocyanin release and degradation [53]. These data suggested that intestinal and colonic microbial digestion of PFSP leads to an accession dependent pattern for anthocyanin accessibility and degradation. Moreover, clinical research plus in vitro and in vivo experiments demonstrated that anthocyanins have preventive and important positive effects on cardioprotection, neuroprotection, and antiobesity as also on antidiabetes and anticancer effects. This is likely due to the rapid absorption of ACNs such that they are seen in the bloodstream within a few minutes of consumption. It is hoped that more detailed research in the future may enable understanding the mechanisms involved whereby food components achieve such effects [54].

### 2.4. Metabolite Diversity

Secondary plant metabolites are essential for performing a variety of cellular functions essential for physiological processes including stress tolerance in plants and likely act as therapeutic agents for betterment of human health. These secondary metabolites are also a source of numerous medicinal compounds while their chemical structure diversity have made them also beneficial for treating serious diseases [55,56,57].

Metabolite profiling of selected 19 pigmented potato cultivars with different anthocyanin profiles in flesh (6 groups) and peel (18 groups) identified 21 anthocyanins including two new anthocyanins in potato (pelargonidin feruloyl-xylosyl-glucosyl-galactoside; cyanidin 3-p-coumaroylrutinoside-5-glucoside) [58]. Significant negative correlation between the main anthocyanins of red and blue potato cultivars was detected for some of the metabolites. This difference in anthocyanin profiles between red and blue potato cultivars possibly relates to regulation of the expression and activities of hydroxylases and methyltransferases rather than to differences for downstream glycosyl- and acyltransferases. Thus, such characterized cultivars represent a valuable genetic resource for the breeding and genetic research of potato anthocyanins [58].

Metabolite profiling of 27 sweet potato accessions including landraces and cultivars involving leaves and storage root tissues unraveled 130 metabolites (secondary metabolism including phenylpropanoids and carotenoids) [59] and lack of correlation between storage root phenotype and leaf metabolism. The accessions differing in root color (yellow, orange, purple) were also significantly different in primary metabolism [59]. Analysis of untargeted metabolomics of the purple-fleshed and one orange-fleshed sweet potato roots of three cultivars on average yielded ~800 mg anthocyanins 100 g^−1^ dry weight. It unfolded mostly acylated peonidin and cyanidin derivatives with ~90% of the total anthocyanin signal, and over 350 flavonoid peaks with variable distribution [60]. Thus, metabolite profiling of germplasm provides an opportunity to identify differences in phenotypes/chemotypes and large structural diversity of anthocyanins and flavonoids in sweet potato.

An entire biosynthetic framework of polyphenol biosynthesis was unfolded via detailed annotation of metabolites and structural genes across *S. lycopersicum* and related species [61]. Identifying such large chemical diversity of polyphenolic compounds and their multiple physiological roles conferring beneficial traits is expected to accelerate omics- and metabolomics-assisted breeding to develop tailor-made tomato cultivars with specific flavonoids.

## 3. Genetic and Molecular Basis of Variation

### 3.1. Anthocyanin Biosynthesis

Anthocyanins constitute a ubiquitous part of flavonoids, a group of polyphenolic molecules in plants of which 600 are known [62]. They also protect plants against pathogens and radiation from UV [63,64]. Common anthocyanin derivatives include cyanidin, delphinidin, malvidin, pelargonidin, peonidin, and petunidin [65]. Anthocyanins bring different and attractive colors to bear on their properties as well as their human health benefits [66,67]. Acetylation, glycosylation, and methylation of anthocyanidins is commonly found in plants and these processes are responsible for anthocyanin diversity [65]. Glycosylation of anthocyanins is catalyzed by glycosyltransferases and these forms are stable, located in vacuoles where they attract pollinators. Similarly, methylation of anthocyanins enables their stability and water solubility, in addition to their accumulation [68]. Non-methylated anthocyanidins, delphinidin, cyanidin, and pelargonidin glycosides contribute to 80% pigmented leaf, 69% fruit, and 50% flowers [69]. However, anthocyanidin modification is important for storage stability of pigmented anthocyanin [70,71] and takes place in the cytosol as soon as anthocyanindin is made [72,73].

Glycosylation at the anthocyanidin C-3 position of a hydroxyl group normally stabilizes anthocyanindins, increasing their hydrophilicity, and stabilizes their color linked to their glycosyl groups [71]. Glycosyltransferases catalyze *0*-glycosylation of anthocyanidin/anthocyanin recognizing UDP-sugar donors such as arabinose, galactose, glucuronic acid, glucose, xylose, and rhamnose.

Overexpression of anthocyanin acyltransferase(s) (AATs) has a great potential in enhancing the color stability of heterologous plants and result in color-modified flowers or plants [74]. DkMYB14 is a transcriptional activator/repressor that directly represses proanthocyanindin (PA) biosynthesis and promotes PA non-solubilization resulting in non-astringent persimmon [75]. On the other hand, MPK4 mediates MYB1 phosphorylation that results in induced anthocyanin accumulation [76]. It is also known that biosynthesis of anthocyanins is activated by a specific protein complex known as MBW (for MYB-bHLH-WDR), which is highly conserved in most flowering plants [77]. In recent years, it has become evident that plants also harbor repressors of anthocyanin, for instance, proteins such as ITCP, HD-ZIP and NAC, which repress or destabilize MBW complex [78].

A huge number of phyto-flavonoids –which include polyphenols, phenolic acids and flavonoids—are now known. Anthocyanins are part of the largest group of plant phenols called flavonoids [43]. In the cultivated tomato fruit, flavonoids including anthocyanins are suboptimal or absent. Examples of the vegetative tissue of solanaceous plants enriched in anthocyanins include pepper [*Capsicum annuum*] (Figure 1A), eggplant (Figure 1B), tomato (Figure 1C), and potato tubers (Figure 1D,E) [79]. Biosynthesis of anthocyanin pathway is well characterized and conserved in plants [66,80,81]. This pathway initiates from 4-coumaroyl-CaO and malonyl COA catalyzed by the enzyme chalcone synthase (CHS) to synthesize naringenin chalcone which is then converted to naringenin by chalcone isomerase (CHI) (Figure 2a). Then, in a series of reactions, naringenin is converted to dihydrokaempferol, catalyzed by flavanonone 3′-hydroxylase (F3H). Flavonoid 3′ -hydroxylase (F3′H) thereafter hydroxylates dihydrokaempferol (DHK) into dihydroquercitin (DHQ) or to dihydro-myricetin (DHM) catalyzed by flavonoid 3′,5′-hydroxylase3′5′H. All three dihydroflavonols DHQ, DHM and DHQ are independently converted step wise to colorless leucoanthocyanidins by the enzyme dihydroflavonol 4-reductase (DFR). The next enzymatic reaction involves the enzyme anthocyanidin synthase (ANS), which converts all three leucoathocyanidins to colored anthocyanidins. Glucosides of peonidin, cyanidin, petunidin and malvinidin are shown in Figure 2b. Finally, the glycosyltransferases attach sugar molecules to the anthocyanidins.

### 3.2. Functionally Characterized Genes Associated with Anthocyanins

Understanding the genetic and molecular basis and identification of genetic tags with trait expression offers great promise to enhance crop breeding efficiency. These genetic tags are defined as strings of DNA that can be used to identify novel allelic variation associated with agriculturally beneficial traits. A candidate gene is a gene identified by its position in a chromosome that relates to trait expression. Such candidates need to be, however, functionally validated prior to being deployed in crop breeding. Here, we focus on functionally characterized genes and recently discovered “candidate” genes associated with anthocyanins.

Genome-wide association analysis involving single nucleotide polymorphisms (SNPs) and diploid potato germplasm of *Solanum tuberosum* L. Group Phureja from Colombia unraveled seven quantitative trait loci (QTL) associated with variation in anthocyanin content. QTL on chromosomes 1 and 10 were found to be the most stable. Three of the seven QTL were similar to previously reported genes involved in the regulation of anthocyanins in potato tubers [37]. In a diverse collection of native landraces of diploid potatoes, phenylalanine-ammonia lyase (*PAL*) gene was contained in the region on chromosome 10, which also harbors other significant SNPs as well as multiple anthocyanin homologs, and is associated with five anthocyanin compounds. *PAL* being pleiotropic is an excellent target for breeding programs as it repeatedly produces significant differences in anthocyanins by recurrent selection. The short distance between *PAL* and multiple MYB transcription factors (TFs) linked with anthocyanin accumulation in potato demonstrated that loci identified in QTL mapping may harbor multiple causal genes. Most potatoes with high anthocyanin content share the same genotype at this cluster, which suggests presence of recurrent selection on the same alleles. Thus, this region has high diversity, consistent with balancing artificial selection to breed potatoes with diverse colors [82].

*StAN1* (*ANTHOCYANIN1*), located on chromosome 10, is a major regulatory gene that controls anthocyanin biosynthesis in potato tubers. The currently available diagnostic markers for identifying functional *StAN1* alleles (*StAN1777*, *StAN1816*) are not efficient to predict potato pigmentation patterns. Thus, there is a need for developing additional diagnostic markers for anthocyanin synthesis that can be used efficiently in potato breeding [83]. This lack of anthocyanins in some potato genotypes may be due to mutation of a structural gene as was evidenced in the case of *StF3*’*5*’, which partially disrupts anthocyanin synthesis that does not affect red but only blue and purple pigmentation. Thus, it is an attractive target for marker-aided identification of potatoes with purple or red flesh tubers [84].

In sweet potato, 22 of the 156 *MYB*-like genes were highly positively or negatively correlated with the anthocyanin content in leaves or storage roots. *IbMYB1* is one of the major anthocyanin biosynthesis regulatory genes. *IbMYB1* constitutes three members, *IbMYB1-1*, *IbMYB1-2a*, and *IbMYB1-2b*, of which, the latter two are not necessary for anthocyanins accumulation in cultivated species that have purple shoots or purplish rings or spots of flesh. The persistent and vigorous expression of *IbMYB1* is essential to maintain the purple color of leaves and storage roots. It is an early response gene of anthocyanin biosynthesis. *IbMYB2s*, the highest similarity genes of *IbMYB1*, are not the members of *IbMYB1*, thereby suggesting there may be more members of *IbMYB1* involved in the regulation of anthocyanin biosynthesis [85]. A genome wide association study (GWAS)-eQTL analysis revealed *IbMYB1-2* as a master regulator and a major gene responsible for the activation of anthocyanin biosynthesis in the storage roots of sweet potato [86].

A novel complex of TFs (namely, IbEFR71-IbMYB340-IbbHLH2), coregulate anthocyanin biosynthesis by binding to the IbANS1 promoter in purple-fleshed sweet potato evidenced by higher expression levels of IbERF71, IbMYB340 and IbbHLH2 in purple-fleshed sweet potatoes. The expression levels were positively correlated with anthocyanin contents. Co-transformation of IbMYB340-IbbHLH2 resulted in anthocyanin accumulation in tobacco leaves and strawberry receptacles, while the addition of IbERF71 significantly intensified the color. TFs together significantly increased the expression levels of *FvANS* and *FvGST*, which are involved in anthocyanin biosynthesis and transport in strawberry, respectively. Thus, work with this crop unfolded a new regulatory network of anthocyanin biosynthesis in purple-fleshed sweet potatoes [87].

*Arabidopsis* seedlings overexpressing *IbWD40* accumulate anthocyanins [88]. Expression of *IbWD40* plays a role in the regulation of anthocyanin biosynthesis in purple sweet potato. Comparative analysis of transcriptome between anthocyanin-containing and anthocyanin-free sweet potato lines unfolded 2329 unigenes that were differentially expressed, of which 1235 were up-regulated and 1094 were down-regulated. The up-regulated unigenes were enriched in the flavonoid and phenylpropanoid biosynthesis pathways, while the down-regulated unigenes were enriched in plant hormone GA signal transduction pathway. The gene co-expression network analysis of differentially expressed unigenes showed that anthocyanin biosynthesis genes were co-expressed with the transcripts related to MYB, bHLH and WRKY TFs [89].

*Solanum chilense Anthocyanin fruit* (*AftAft/-*) and *atroviolacea* (*atvatv*) loci contribute to increased anthocyanin accumulation in fruits of cultivated tomato [90,91,92]. The CRISPR/Cas9-mediated *Slan2* mutants had fruit color and anthocyanin content similar to the wild type (WT) cv. ‘Indigo Rose’. However, in the *SlAN2* mutant, the anthocyanin content and relative expression levels of anthocyanin-related genes were significantly lower in vegetative tissues relative to WT [93]. Another putative candidate gene *Slan2-like* in the *Aft* locus fine mapped to an approximately 145-kb interval on chromosome 10. The CRISPR/Cas9-mediated *Slan2-like* mutants in comparison to WT showed much lower anthocyanins due to the downregulation of multiple anthocyanin genes. This suggests that *Slan2-like* may be a good candidate gene to enhance the anthocyanin content in tomato [94]. Alternative splicing is “a process that allows the generation of different forms of mature mRNA from the primary gene transcript” [95]. A study on comparative functional analysis of R2R3 MYB transcription factors in wild-type and *Aft* plants demonstrated significant differences both in the expression level and protein functionality of *Slan2like*. Splicing mutations resulted in a complete loss of function of the WT protein, thereby indicating its key role in the anthocyanin pigmentation in tomato fruit [96]. The photomorphogenesis-related TF SIBBX20 regulates anthocyanin accumulation in tomato [97]. SIBBX20 promotes anthocyanin biosynthesis by binding the promoter of the anthocyanin biosynthesis gene *SIDFR*. SIBBX20 interacts with COP9 signalosome subunit SICSN5-2. Silencing of *SICSN5-2* led to anthocyanin hyperaccumulation in transgenic tomato calli and shoots, while *SICSN5-2* overexpression decreased anthocyanin accumulation, thereby suggesting that SIBBX20-SICSN5 modulation may represent a novel regulatory pathway in light-induced anthocyanin biosynthesis in tomato [98].

Significant progress has been made in identifying genes associated with anthocyanin biosynthesis in food crops, some of which have been functionally characterized (Table 1). However, a focused approach is needed to functionally characterize other “candidate” genes to ascertain their involvement in anthocyanin biosynthesis and to develop genetic tags for use as aids in crossbreeding.

## 4. Biofortification to Redesign Next-Generation Anthocyanins Rich Vegetable Crops

Biofortification refers to the development of nutrient- or phytochemical-dense crops using crossbreeding or genetic engineering. It is a sustainable and cost-effective strategy to address all forms of malnutrition, particularly in the developing world. Crop biofortification has been successful in developing and releasing micronutrients (Fe and Zn) or provitamin A dense grain and root (orange-fleshed sweet potato, OFSP) crops across continents [99,100]. Micronutrient deficiency is widespread in the Africa, Asia, and Latin America, and regular consumption of biofortified crops has been shown to increase micronutrient intakes and thus help meet the sustainable development goals [101]. Integrating high seed nutrients (Fe, Zn) density with greater pro vitamin A and flavonoids (anthocyanin) as strategy in breeding program may provide lasting solution to overcome all forms of malnutrition in developing economies.

### 4.1. Crossbreeding and Genomic-Assisted Breeding

A few anthocyanin-rich potato and sweet potato cultivars have been released for cultivation in India. A new specialty table potato cultivar named ’Kufri Neelkanth’ ensued from crossbreeding (https://icar.org.in/content/kufri-neelkanth-new-antioxidant-rich-potato-variety-developed-icar-central-potato-research, accessed on 22 January 2022). The average tuber yield of ’Kufri Neelkanth’ ranges from 35 to 38 t ha^−1^, which is much higher than the Indian national average of 23 t ha^−1^. It is speculated that once this cultivar is commercially grown, it would be a boon to farmers in the northern Indian plains, in particular Punjab where ~2.7 million t of annual potato production is known and 60 to 70% of seed potato is distributed to domestic markets https://www.potatopro.com/news/2019/newly-bred-indian-potato-variety-said-be-rich-antioxidants-ready-commercial-production, accessed on 22 January 2022).The sweet potato ‘Bhu Krishna’ has been bred in India [102] while sweet potato cultivar ‘P 4′, potato cultivars ‘AmaRosa’, ‘Purple Pelisse’, and ‘Terra Rossa’ have been bred in the USA [103,104,105,106]. The colored potato cultivars are widely cultivated in Europe, South America, North America, and to a lesser extent in Asia [14,75]. Anthocyanin-rich (0.04 to 0.12 mg g^−1^ fresh tuber weight) advanced CIP potato clones, including CIP302281.25, CIP302302.34, CIP302299.16, CIP302288.35, CIP302281.39, CIP302298.44, CIP302298.16, CIP302304.27, and CIP302306.33, are being globally utilized in potato breeding for developing locally-adapted anthocyanin-rich cultivars (https://research.cip.cgiar.org/cipcatlg_ac/breeders.php?language=1&name=English, accessed on 22 January 2022).

Genetic engineering has become an ideal tool to develop anthocyanin-rich tomatoes [48,96]. Some wild tomato species, such as *S. chilense* and *S. cheesmaniae*, biosynthesize anthocyanins in the fruit sub-epidermal tissue, and some alleles from those genotypes have been introgressed into a newly developed purple tomato line named ‘Sun Black’ (SB). This tomato cultivar has a deep purple skin color due to higher anthocyanins in the peel, and a normal red-colored pulp and taste similar to traditional tomato has been released in Italy utilizing a crossbreeding approach [107]. Notably, a blue-colored tomato with high anthocyanin levels had been released earlier in the USA [108]. A breeding line with purple fruit color was bred by involving in its development ‘OSU blue’ (blue fruit) and ‘Purple mini’ (brown fruit). In this line, both the early and late biosynthesis genes (except *Sl5GT*) related to anthocyanin biosynthesis are upregulated in the peels of purple tomato fruits. The expression of regulatory genes *SlANT1* and *SlAN1* dramatically increased in the peels of purple tomato fruits, thereby suggesting that both genes control anthocyanin biosynthesis in the peels of purple-fruited tomatoes via the up-regulation of structural genes in the anthocyanin pathway. It is therefore an important genetic resource for use in potato breeding [109].

An introgression line carrying anthocyanin biosynthesis regulatory genes (*Aft*, *atv*, *hp2*) in the genetic background of red-fruited tomato showed enhanced accumulation of anthocyanins and bioactive compounds and distinct changes in volatile compounds [110]. Stacking of lycopene (*dg*) and anthocyanin (*Aft*) synthesizing genes led to selection of a purple-fruited tomato (*AftAftdgdg*) with exceptionally high levels of lycopene (~6 mg 100 g^−1^ fresh weight, FW), anthocyanin (~21 mg 100g^−1^ FW), and ascorbic acid (~31 mg^−1^ FW) [111]. Interestingly, a red-fruited tomato line was converted into a purple-fruited commercial cherry line LAM374 stacking *Aft*, *atv*, and *hp2* alleles [112].

Significant progress has been made in developing anthocyanins-rich potato, sweet potato, and tomato cultivars, advanced breeding lines, and genetic stocks (Table 2). Combining high anthocyanins into improved genetic backgrounds is a significant breeding challenge [113]. A more directed effort is needed to recycle the currently available anthocyanins-rich gene pools in crop breeding to increase yield ceilings, especially among anthocyanin-rich grain crops. However, phenolic compounds including high levels of anthocyanin can diminish iron absorption from anthocyanin rich vegetables [114].

### 4.2. Transgenes and CRISPR/Cas9 Systems to Enrich Anthocyanins

Genetic engineering of anthocyanin biosynthesis pathway in staple crops can provide health promoting food in abundance to minimize the risk of NCDs in humans. As stated above, anthocyanin biosynthesis pathway is complex and regulated by complex genetic network that involves both structural and regulatory genes. However, success has already been achieved in enhancing anthocyanin levels in potato, sweet potato and tomato with little or no adverse impact on plant growth and development. Anthocyanin biosynthesis regulation is fairly conserved among plants; however, different isoforms of anthocyanin-related genes between cultivars results in tissue-specific accumulations of purple pigments. The key targets identified for developing the anthocyanin-rich cultivars through genomic-assisted breeding approaches include MYB-bHLH-WD40 TFs [121,122,123]. *Leaf Color (Lc)* TF gene enabled sweet potato transformation increasing both content (1.5–1.9 times compared to the respective WT) and composition (17 anthocyanins in WT and 19 in transgenic lines), while some individual anthocyanins increased by 4.5 times and others decreased by ~2 times [124].

Transgenic tomato plants modified using *Delila* and *Rosea1* genes from snapdragon (*Antirrhinum majus*) were enriched with a 70 to 100-fold increase in anthocyanin content in purple fruit [47]. These transgenic plants had no negative effects on other quality attributes such as total carotenoids, including lycopene levels. Similarly, *Delila* and *Rosea1* genes from transgenic tomato cultivar ‘Micro-Tom’ were transferred to tomato cultivar ‘Moneymaker’ by crossbreeding which resulted in the enhancement of fruit anthocyanin by 131% (3 g kg^−1^ dry matter) [125]. In addition to these studies, when the *SlMYB75* gene was overexpressed in tomato fruit, multiple quality traits were improved, e.g., anthocyanin content reached 1.86 mg g^−1^ fresh weight and total phenolics, flavonoids and soluble solids relative to WT increased by 2.6-, 4-, and 1.2-fold, respectively [126]. Also, the aroma volatiles (aldehyde, phenylpropanoid-derived and terpene volatiles) significantly increased in the transgenic tomato fruit as compared to WT. The transcript profiling of these transgenic tomatoes revealed that the genes involved in ethylene signaling, phenylpropanoid and isoprenoid pathways were upregulated, thus suggesting that *SlMYB75* is a key regulator of fruit quality attributes [126].

### 4.3. Biosafety Regulation and Acceptability of Genetically Modified Anthocyanin-Rich Vegetables

The transgenic “anti-cancer” purple tomato—expressing two snap dragon genes with increased anthocyanin content—captured the attention of potential end-users due to its proposed human health benefit [45,47]. Some grocery shops are reluctant to sell the transgenic anthocyanin-rich tomatoes because they think consumers may reject such produce, and their imagined concern about the safety of transgenic-derived food [127,128]. Biosafety assessment, however, indicated that carotenoid levels did not change in the anthocyanin-rich lines, thus indicating the stability of the transgene [33]. It is now considered by the US Food and Drug Administration (FDA) that plant-edited produce can be cultivated and sold with no regulation.

## 5. Concluding Remarks

Secondary metabolism in plants is a source of several potential bioactive compounds some of which have been linked to human health. Anthocyanins belong to a major group of polyphenolic molecules called flavonoids. The biosynthetic pathway(s) leading to synthesis of these bioactive compounds is (are) regulated by a complex of structural and regulatory genes as well as environmental cues. Hence, anthocyanin biofortification will require new knowledge and advancements in gene stacking through crossbreeding programs and metabolic engineering to ensure rich germplasm for nutrition needs of growing human population particularly in the developing world. Such advancements may also ensure that commodities deficient in anthocyanins, such as a commonly bred tomato, can generate a good dose of nutritional molecules. Notably, genetic engineering technology successfully led to the enhancement of the phytonutrient content of the anti-cancer carotenoid lycopene (by 200–300%), juice quality, and vine life of tomato fruit [44]. Moreover, as discussed above, engineering of transcription factors and gene editing technology were successfully applied in developing anthocyanin-rich food crops without any significant yield penalty. Biofortification (of both micronutrients and phytochemicals) utilizing metabolic engineering can potentially eradicate malnutrition and thereby enhance global human health [129]. An approach to eating a balanced diet, saving land area, and reducing greenhouse gas emissions would require more production and consumption of fruits and vegetables and diets high in desirable proteins [130]. Public policy targeting the constraints to produce and consumption of fruits and vegetables is needed. A portfolio of interventions and investments that focus on fruit and vegetable production, technologies and practices to reduce waste without increasing consumer cost, and educating the public in large on healthy and tested diets [131].

## Figures and Tables

**Figure 1 ijms-23-02634-f001:**
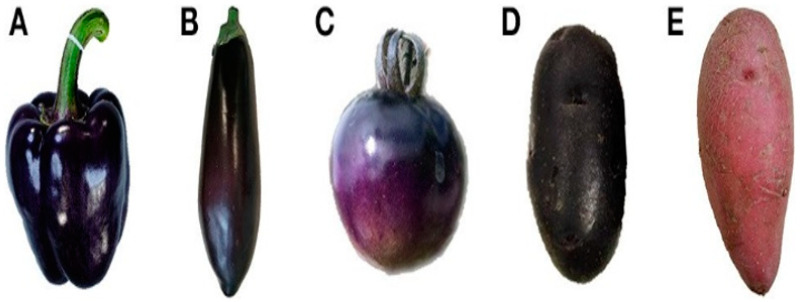
S*olanaceous* vegetables rich in anthocyanins: (**A**) purple pepper fruit, (**B**) purple eggplant fruit, (**C**) purple tomato fruit, (**D**) purple potato tuber, (**E**) red potato tuber. (After Liu et al. [66]; thanks to *Frontiers in Chemistry*).

**Figure 2 ijms-23-02634-f002:**
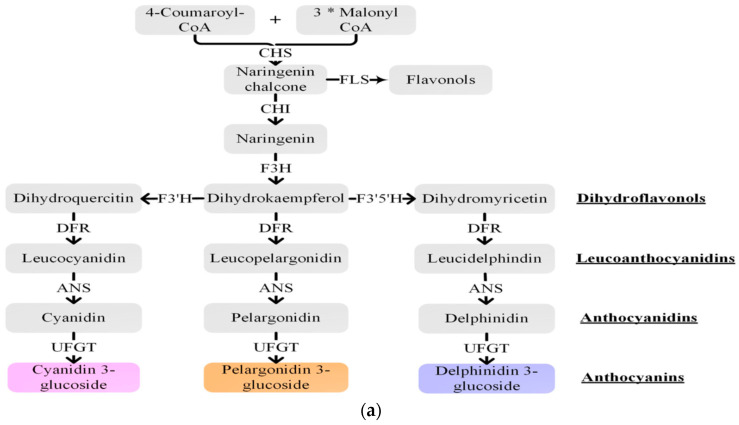
(**a**). Schematic representation of the anthocyanin biosynthetic pathway. CHS, chalcone synthase; CHI, chalcone isomerase; F3H, flavanone 3-hydroxylase; F3′H, flavonoid 3′-hydroxylase; F3′5′H, flavonoid 3′,5′-hydroxylase; DFR, dihydroflavonol 4-reductase; ANS, anthocyanidin synthase; UFGT, flavonoid 3-O-glucosyltransferase; FLS, flavonol synthase. The “*” means multiplication. After Liu et al. [66] (thanks to *Frontiers in Chemistry*). (**b**). Glucosides of other anthocyanidins. Shown are peonidin made from cyanidin while delphinidin is converted to petunidin, which in turn is converted to malvinidin (thanks to Dr. Tahira Fatima for help with this figure. See above text for details).

**Table 1 ijms-23-02634-t001:** Candidate or functionally characterized genes regulating anthocyanin biosynthesis in potato, sweet potato, and tomato.

Gene	Description	Reference
**Potato**		
*PAL*	*PAL*, being in close proximity of multiple MYB TFs on chromosome 10, linked with anthocyanin accumulation; it being pleiotropic an excellent target for enhancing anthocyanin content by recurrent selection	[82]
*StF3*’*5*’	Partially disrupt anthocyanin synthesis affecting blue and purple but not red pigmentation; attractive target for marker-aided identification of potatoes with purple or red flesh color tubers	[84]
**Sweet Potato**		
*IbMYB1*	A major anthocyanin biosynthesis regulatory gene	[85]
*IbANS1*	IbERF71-IbMYB340-IbbHLH2, a novel TF complex, coregulate anthocyanin biosynthesis by binding to the IbANS1 promoter in purple-fleshed sweet potatoes than other color cultivars; expression levels positively correlated with anthocyanin contents	[87]
*IbMYB1*, *IbWD40*	*IbMYB1* expressed in purple-fleshed cultivars but not in other with orange-, yellow-, or white-fleshed color; *IbWD40* expression limited to one anthocyanin rich cultivar; Arabidopsis seedling overexpressing *IbWD40* accumulated anthocyanins, indicating that it regulates anthocyanin biosynthesis in purple sweet potato	[88]
**Tomato**		
*SIDFR*	SIBBX20 promotes anthocyanin biosynthesis by binding the promoter of the anthocyanin biosynthesis gene *SIDFR*	[98]
*Slan2*	*Slan2* mutants regulate anthocyanins in *Aft* locus	[93]
*Slan2-like*	*Slan2-like* mutants downregulate anthocyanins in *Aft* locus	[94]

**Table 2 ijms-23-02634-t002:** List of anthocyanins-rich potato, sweet potato and tomato cultivars/advanced breeding lines and genetic stocks developed through crossbreeding and selection.

Cultivar	Characteristics	Reference
**Potato**		
‘Kufri Neelkanth’	Adapted for cultivation in northern Indian plains; much higher yield (35 to 38 t ha^−1^) compared to national average of 23 t ha^−1^; anthocyanin >100 µg 100 g^−1^ fresh weight (FW) [115]	https://icar.org.in/content/kufri-neelkanth-new-antioxidant-rich-potato-variety-developed-icar-central-potato-research, accessed on 22 February 2022
‘Puma Makin’, ‘Leona’, ‘Yawar Manto’, ‘Añil’, ‘Sangre de Toro’, ‘Qequrani’	Native pigmented (red and purple) cultivars grown in Huancavelica region of Peru; Anthocyanin: ‘Puma Makin’, 74.3 mg kg^−1^ skin dry weight, SDW; Leona, 6.32 mg kg^−1^ flesh dry weight, FDW and 166.65 mg kg^−1^ SDW; ‘Yawar Manto’, 602.9 mg kg^−1^ FDW and 709.4 mg kg^−1^ SDW; ‘Añil’, 104 mg kg^−1^ FDW and 273.9 mg kg^−1^ SDW; ‘Sangre de Toro’, 27.5 mg kg^−1^ FDW and 124.2 mg kg^−1^ SDW; ‘Qequrani’, 10.2 mg kg^−1^ FDW	[32]
‘AmaRosa’, ‘Purple Pelisse’, and ‘Terra Rossa’	Released for cultivation in USA (‘AmaRosa’: Total anthocyanin content (TAC) 18.2 mg compared to 13.8 mg 100 g FW in control ‘All Blue’; ‘Purple Pelisse’: TAC 34.2 mg compared to 12.6 mg 100 g FW in control ‘All Blue’)	[104,105,106]
‘Hongyoung’, ‘Jayoung’	Released for cultivation in South Korea (‘Hongyoung’: TAC 31.8 mg 100 g FW, 3.6 times higher than control Jasim)	[116,117]
‘Hermanns Blaue’, ‘Vitelotte’, ‘Shetland Black’, ‘Valfi’	European blue-fleshed potato cultivars	[118]
**Sweet Potato**		
‘Antin 1’, ‘Antin 2’, ‘Antin 3’	Released in Indonesia (TAC 8.4, 130.2, and 150.7 mg 100 g^−1^ FW)	[26]
‘Bhu Krishna’	Released in India (TAC 90.0 mg 100 g compared to nil in control)	[102]
P4	Released in USA (TAC up to 14 mg g DW)	[103]
‘Borami’, ‘Mokpo 62’; ‘Shinzami’, ‘Zami’	Released in South Korea	[119]
‘Yamagawamurasaki’, ‘Ayamurasaki’, ‘Chiran Murasaki’, ‘Tanegashima Murasaki’, ‘Naka Murasaki’, ‘Purple Sweet’	Released for cultivation in Japan	[120]
**Tomato**		
Near isogenic line (NIL)	Enhanced accumulation of anthocyanins and bioactive compounds with distinct changes in volatile compounds in NIL carrying *Aft*, *atv*, and *hp2* in the genetic background of red-fruited tomato	[110]
Breeding line	A purple-fruited line derived from a cross between ‘OSU blue’ (blue fruit) and ‘Purple mini’ (brown fruit)	[109]
‘Sun Black’	Deep purple skin but with a normal red color pulp and taste similar to traditional tomato released in Europe; TAC 7.1 mg 100 g FW, comparable to eggplant or red cherry [48]	[107]
‘Indigo Rose’	A blue colored tomato released for cultivation in USA (TAC up to 10 mg 100 g^−1^ FW on a whole fruit basis, normal tomato fruits devoid of anthocyanin)	[108]

## Data Availability

Not applicable.

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
