# Peer review of "Anthocyanin-Rich Vegetables for Human Consumption—Focus on Potato, Sweetpotato and Tomato"

_ijms, 2022, doi:10.3390/ijms23052634_

Round 1
Reviewer 1 Report
The manuscript entitled “Anthocyanin-Rich Vegetables for Human Consumption – Focus on Potato, Sweet potato and Tomato”, authored by Mattoo and colleagues, deals with the reviewing of genetic variation harnessed through crossbreeding and biotechnology-led approaches for developing anthocyanins-rich fruit and vegetable crops. Anthocyanins are very important bioactive compounds, both from a plant and a human point of view, due to their bioactivity and their potential to regulate transcriptional and post transcriptional pathways. Recently, these compounds are gaining more and more importance especially in human health. For this reason, I believe that the topic of this review is very important and interesting.
The manuscript is very well written, deals with the topics with extreme professionalism, and contains very important information. However, before the revised manuscript can be judged suitable as publication for IJMS, small comments need to be made.
Despite this reviewer is well aware of the difficulty of writing an abstract with a very limited number of words, this section should be a single paragraph, of a maximum of 200 words. The proposed abstract far exceeds the guidelines provided by the journal. Consequently, I would recommend limiting and shortening this section following the Journal's guidelines.
Keywords should be words not contained in the title or in the abstract. Their usefulness is to make easier the searching of the article using the common scientific search engines. Since several keywords are already present in the title, and/or repeated several times in the abstract, I strongly advise the authors to replace some of them and add more. As journal guidelines clearly report, a limited number of keywords can be used. Consequently, authors should carefully choose them.
The introductory section should also mention that lately several scientific articles, including original papers and reviews, have been published on this same topic (namely anthocyanins). This index should make clear the growing interest for these bioactive compounds, not only from a pharmaceutical level but also from a nutritional and nutraceutical point of view. Indeed, apart from chemical engineering strategies, other ways to increase the anthocyanin content using a “more natural” approach are investigated. For example:
- Use of biostimulant products to increase anthocyanin level in fruits and vegetables: Mannino, G., Gentile, C., Ertani, A., Serio, G., & Bertea, C. M. (2021). Anthocyanins: Biosynthesis, Distribution, Ecological Role, and Use of Biostimulants to Increase Their Content in Plant Foods—A Review. Agriculture, 11(3), 212.
- Use of mycorrhiza to increase anthocyanin content in red rice: Wangiyana, W., Aryana, I. G. P. M., & Dulur, N. W. D. (2021). Mycorrhiza biofertilizer and intercropping with soybean increase anthocyanin contents and yield of upland red rice under aerobic irrigation systems. In IOP Conference Series: Earth and Environmental Science (Vol. 637, No. 1, p. 012087). IOP Publishing.
- Use of semi-natural compounds (such as melatonin or phytomelatonin) to increase anthocyanin accumulation in red pear: SUN, Hui-li, et al. "Preharvest application of melatonin induces anthocyanin accumulation and related gene upregulation in red pear (Pyrus ussuriensis)." Journal of Integrative Agriculture 20.8 (2021): 2126-2137.
In Section 2.2. the authors speculate about the anthocyanin content in different tomato and potato varieties. One of the aforementioned articles related to anthocyanins report a cluster analysis in which different plant species were classified based on the total anthocyanin content. Authors should refer to this article and compare their data with those of other species well known to contain anthocyanins
(DOI: 10.3390/agriculture11030212).
Author Response
|
The manuscript entitled “Anthocyanin-Rich Vegetables for Human Consumption – Focus on Potato, Sweet potato and Tomato”, authored by Mattoo and colleagues, deals with the reviewing of genetic variation harnessed through crossbreeding and biotechnology-led approaches for developing anthocyanins-rich fruit and vegetable crops. Anthocyanins are very important bioactive compounds, both from a plant and a human point of view, due to their bioactivity and their potential to regulate transcriptional and post transcriptional pathways. Recently, these compounds are gaining more and more importance especially in human health. For this reason, I believe that the topic of this review is very important and interesting. The manuscript is very well written, deals with the topics with extreme professionalism, and contains very important information. However, before the revised manuscript can be judged suitable as publication for IJMS, small comments need to be made. |
|
|
Despite this reviewer is well aware of the difficulty of writing an abstract with a very limited number of words, this section should be a single paragraph, of a maximum of 200 words. The proposed abstract far exceeds the guidelines provided by the journal. Consequently, I would recommend limiting and shortening this section following the Journal's guidelines. |
Condensed <200 words (lines 12-23) |
|
Keywords should be words not contained in the title or in the abstract. Their usefulness is to make easier the searching of the article using the common scientific search engines. Since several keywords are already present in the title, and/or repeated several times in the abstract, I strongly advise the authors to replace some of them and add more. As journal guidelines clearly report, a limited number of keywords can be used. Consequently, authors should carefully choose them. |
Revised as per guidelines (line 24) |
|
The introductory section should also mention that lately several scientific articles, including original papers and reviews, have been published on this same topic (namely anthocyanins). This index should make clear the growing interest for these bioactive compounds, not only from a pharmaceutical level but also from a nutritional and nutraceutical point of view. |
Added (lines 53-55) |
|
Indeed, apart from chemical engineering strategies, other ways to increase the anthocyanin content using a “more natural” approach are investigated. For example:
|
Cited (lines 55-57) |
|
In Section 2.2. the authors speculate about the anthocyanin content in different tomato and potato varieties. One of the aforementioned articles related to anthocyanins report a cluster analysis in which different plant species were classified based on the total anthocyanin content. Authors should refer to this article and compare their data with those of other species well known to contain anthocyanins |
Cited (lines 96-104; 129-132; 150-151) |

Reviewer 2 Report
The review included sections about improved accumulation of bioactive compounds, specifically anthocyanins and also about its the metabolic pathway. However information about sources of variation in the anthocyanin levels (like effect of environment) are missing and can be incorporated.
Since this is a review of anthocyanins under human consumption perspective it is important to express the levels of total and individual anthocyanins in fresh weight. Given the fact that potatoes and sweetpotatoes are eaten as cooked, I also suggest including a section under genetic variability focusing in cooked samples and also regarding biotransformation after consumption
Further it will be good to include a discussion comparing the total and individual anthocyanin of levels of anthocyanins rich potato, sweet potato, and tomato cultivars to the anthocyanin concentrations in varieties commonly consumed in the country where the variety has been released and also a comparison with other rich anthocyanin food.
In section 4.1. Crossbreeding and genomic-assisted breeding discuss about the developed anthocyanin rich varieties, would be interesting to mention how rich are the improved potato varieties compared with the most common consumed varieties in the area where they have been released. I suggest to add in table 2 a column for the improved level of anthocyanins and a column with the levels in the most consumed varieties.
The concluding remarks mention biofortification but biofortification has not been mentioned before in the review. I suggest to include a section about biofortification for high anthocyanin levels emphasizing that high anthocyanin levels will improve the health promoting effect of potatoes. However it is also important to mention that phenolics compounds including high levels of anthocyanin can diminish iron absorption from potatoes (Jongstra et al., 2020). https://pubmed.ncbi.nlm.nih.gov/33188398
Author Response
|
The review included sections about improved accumulation of bioactive compounds, specifically anthocyanins and also about its the metabolic pathway. However information about sources of variation in the anthocyanin levels (like effect of environment) are missing and can be incorporated. |
Added (lines 81-93) |
|
Since this is a review of anthocyanins under human consumption perspective it is important to express the levels of total and individual anthocyanins in fresh weight. |
Ref table 2 (wherever available, data on total anthocyanin content, TAC cited), while individual anthocyanin data not available in cited literature across cultivars/germplasm |
|
Given the fact that potatoes and sweetpotatoes are eaten as cooked, I also suggest including a section under genetic variability focusing in cooked samples and also regarding biotransformation after consumption |
Added: anthocyanin variation on cooked samples in germplasm (lines 133-135), & on biotransformation (lines 168-176) |
|
Further it will be good to include a discussion comparing the total and individual anthocyanin of levels of anthocyanins rich potato, sweet potato, and tomato cultivars to the anthocyanin concentrations in varieties commonly consumed in the country where the variety has been released and also a comparison with other rich anthocyanin food. |
Ref table 2 (wherever available, data on TAC in released & control cv cited), while individual anthocyanin data not available in cited literature |
|
In section 4.1. Crossbreeding and genomic-assisted breeding discuss about the developed anthocyanin rich varieties, would be interesting to mention how rich are the improved potato varieties compared with the most common consumed varieties in the area where they have been released. I suggest to add in table 2 a column for the improved level of anthocyanins and a column with the levels in the most consumed varieties. |
Ref table 2 (wherever available, data on TAC in newly developed and control cvs cited), while individual anthocyanin data not available in cited literature |
|
The concluding remarks mention biofortification but biofortification has not been mentioned before in the review. I suggest to include a section about biofortification for high anthocyanin levels emphasizing that high anthocyanin levels will improve the health promoting effect of potatoes. |
Added (lines 338-347) |
|
However it is also important to mention that phenolics compounds including high levels of anthocyanin can diminish iron absorption from potatoes (Jongstra et al., 2020). https://pubmed.ncbi.nlm.nih.gov/33188398 |
Cited (lines386-387) |
